# Adapting Graph-Based Analysis for Knowledge Extraction from Transformer Models

**Alexandre Monnier Weil**            ALEXANDRE.MONNIER@UNIV-GRENOBLE-ALPES.FR
*Université Grenoble Alpes, Saint-Martin-d'Hères, France*

**Vitor A. C. Horta**            VITOR.HORTA@PEERS.COM.BR
*Peers Consulting + Technology, Brazil*

**Hamza Qadeer**        HAMZA.QADEER@INSIGHT-CENTRE.ORG and **Alessandra Mileo**[*]
ALESSANDRA.MILEO@INSIGHT-CENTRE.ORG
*Insight Centre for Data Analytics, Dublin City University, Dublin, Ireland*

**Editors:** Leilani H. Gilpin, Eleonora Giunchiglia, Pascal Hitzler, and Emile van Krieken

## Abstract

Transformer models, despite their exceptional capabilities in Natural Language Processing (NLP) and Vision tasks, like deep neural network models, often function as "black boxes" as their internal processes remain largely opaque due to their complex architectures. This work extends graph-based knowledge extraction techniques, previously applied to CNNs, to the domain of Transformer models. The inner mechanics of Transformer models are explored by constructing a co-activation graph from their encoder layers. The nodes of the graph represent the hidden unit within each encoder layer, while the edges represent the statistical correlations between these hidden units. The magnitude of co-activation, which is the correlation between activations of two hidden units, determines the strength of their connection within the graph. Our research is focused on encoder-only Transformer classifiers. We conducted experiments involving a custom-built Transformer and a pre-trained BERT model for an NLP task. We used graph analysis to detect semantically related class clusters and their impact on misclassification patterns. We demonstrate a positive correlation between class similarity and the frequency of classification errors. Our findings suggest that co-activation graphs reveal structured, interpretable representations in Transformers, consistent with prior CNN findings on knowledge extraction.

## 1. Introduction

Transformer-based architectures Vaswani et al. (2017) have reshaped natural language processing (NLP), significantly enhancing performance in machine translation, sentiment analysis, and question answering Chernyavskiy et al. (2021); Wang et al. (2019).

Moreover, their influence extends beyond NLP into visual tasks such as image classification and object detection, surpassing traditional methodologies while offering a richer interpretability of visual information Khan et al. (2022); Dosovitskiy et al. (2021); Jamil et al. (2022).

Consequently, Transformer models are revolutionizing the realms of both NLP and computer vision. Their extraordinary success is attributed to the self-attention mechanism. Nevertheless, despite their impressive performance across various tasks, these architectures

---

[*] Corresponding author

introduce significant interpretability and explainability challenges. Especially in areas like law enforcement and healthcare, accurate predictions are not sufficient; the models must also elucidate their decision-making process in a human-understandable manner. Numerous methods generate attention-based heatmaps to explain Transformers Abnar and Zuidema (2020); Chefer et al. (2021). However, they often miss the complex interactions among layers and heads, explaining only fragments of the model's behavior.

This work builds upon prior approaches based on CNN-based coactivation graphs and extends it to the Transformer classifiers of encoder-only models such as BERT and ViT by extracting knowledge via co-activation graphs Horta et al. (2021). A co-activation graph links every pair of neurons based on activation-value correlations: nodes represent hidden units, and edges their co-activation strength. This enables graph-based analyses, such as community detection and link prediction, to uncover structured representations in Transformers. Our main contributions are: (i) A method to build co-activation graphs from Transformer encoder layers. (ii) Evidence of a positive correlation between class similarity and classification errors. (iii) Graph-based identification of semantically similar class clusters and their impact on misclassification.

The rest of the paper is organized as follows. Section 2 reviews Transformer architectures and prior CNN-based co-activation graphs. Section 3 describes our co-activation graph methodology. Section 4 presents experiments on NLP classifiers. Additional results on ViT appear in the appendix. Section 5 concludes and outlines future work.

## 2. Related Work

This section reviews the basics of Transformer models, focusing on encoder architecture for classification. It also summarizes existing interpretability methods designed for Transformers and revisits the co-activation approach previously applied to CNNs Horta et al. (2021), as this work builds upon that methodology for Transformer-based architectures.

### 2.1. Transformer Encoder Architecture and Embedding Space

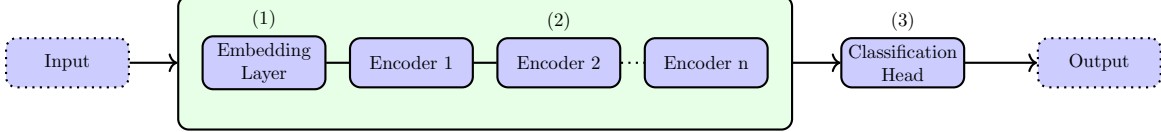

Figure 1: Diagram of a Transformer architecture for classification

Transformers, a widely adopted deep learning model introduced by Vaswani et al. (2017), use attention mechanisms to process sequential data effectively, yielding substantial improvements over prior architectures. They have become foundational in both natural language processing and computer vision tasks.

The original Transformer comprises two components: an encoder and a decoder. While machine translation tasks use both, classification tasks typically require only the encoder, as illustrated in Figure 1. At the core lies the embedding layer (1), which transforms discrete tokens—such as words, sub-words, or characters—into dense vectors within a multi-

dimensional *embedding space.* This space encodes semantic and syntactic information; similar tokens cluster together, and positional encodings preserve word order.

Table 1: Example of values at the end of an encoder

|  | $d_1$ | $d_2$ | $d_3$ | $d_4$ | $d_5$ |
|---|---|---|---|---|---|
| I | -0.6 | 0.1 | 0.9 | -0.3 | -0.7 |
| love | 0.5 | -0.9 | 0.6 | 0.8 | -0.2 |
| co-activation | 0.3 | -0.7 | -0.8 | 0.4 | 0.6 |
| graph | -0.1 | 0.9 | -0.5 | -0.4 | 0.7 |

As these vectors pass through the encoder layers[1] (2), they are refined by multi-head self-attention mechanisms, which model all token relationships in parallel, regardless of distance. Over successive layers, the embeddings evolve into context-rich representations. Table 1 shows the kind of vectors generated by the final encoder layer.

After passing through the final encoder, the representation is processed by a linear layer (3) followed by softmax to produce the class probability distribution.

## 2.2. Explainability for Transformers

Most Transformer explainability efforts rely on attention maps Xu et al. (2016); Carion et al. (2020). However, Chefer et al. (2021) argued that attention weights alone do not capture the full reasoning process. They introduced a relevance scoring method based on deep Taylor decomposition, which propagates through attention layers and skip connections.

Voita et al. (2019) investigated the impact of individual attention heads on model performance, showing that interpretable, specialized heads are typically pruned last using stochastic gates and a relaxed L0 penalty.

Abnar and Zuidema (2020) introduced attention flow and attention deployment to aggregate attention across layers; attention flow enhances traceability but lacks scalability, while attention deployment struggles to distinguish positive from negative contributions. More recently, Marks et al. (2025) proposed *sparse feature circuits*, interpretable causal subnetworks that provide mechanistic insights and improve downstream task performance through targeted feature ablation.

Together, these works highlight that attention-based heatmaps are useful yet limited, often capturing local rather than global model behavior. Our work proposes a graph-based approach for generating structured, global explanations of Transformer models.

## 2.3. Co-activation Graphs for CNNs

To analyze internal representations of CNNs, Horta et al. (2021) introduced the co-activation graph method. We briefly review this approach here.

A co-activation graph is constructed by feeding samples through a network and capturing the activation values. Each neuron becomes a node, and edges are weighted by the statistical correlation (e.g., Spearman coefficient) between neuron activations. For convolutional layers, average pooling condenses multi-region outputs into a single representative

---

1. 'Encoder n' in the diagram denotes that any number of encoders can be stacked.

value. An empirical threshold retains only the strongest connections to simplify graph construction. These graphs enable analysis using community detection, centrality measures, and link prediction Horta and Mileo (2021). Communities often correspond to functionally similar neuron groups and class-level features. Centrality metrics identify influential neurons affecting predictions.

Overall, co-activation graphs serve as both visual tools and analytical structures for interpreting learned representations in neural models.

## 3. Generating co-activation graphs for transformers

The main goal is to interpret the knowledge acquired by Transformers during training. To achieve this, we employ graph methods such as community analysis, which help identify groups of classes that are more similar to each other. However, co-activation graphs were originally proposed only for convolutional and fully connected layers. Thus, before applying these methods to Transformers, we must first devise a way to construct co-activation graphs for encoder layers. This will be developed in this section, and the flow chart is presented in Appendix A for better understanding.

To clarify terminology, we use the term *hidden unit* to refer to the dimensions of the embedding space at the encoder's output.

Constructing a co-activation graph from a Transformer's encoder output poses challenges. The encoder's output matrix corresponds directly to the input sequence, so each element has a context tied to its input token. Directly correlating these elements may not provide meaningful insights, since activations depend heavily on specific inputs. Alternatively, the encoder's output can be viewed as a set of activation vectors, each for a distinct *hidden unit*. Hence, finding meaningful correlations to build a useful co-activation graph is non-trivial.

An effective technique to calculate correlation between these vectors is reducing their dimensionality to a single value via an operation that retains significant information. The mean operation is a good candidate, offering a simple, efficient method that yields a value capturing the core essence of the vector. This consistent approach ensures every data point influences the final representation, preserving relative relationships between vectors. Table 2 illustrates how to compute the average vector for a sequence:

Table 2: Dimension-wise averages for the sequence

|                         | $d_1$  | $d_2$  | $d_3$ | $d_4$  | $d_5$  |
|-------------------------|--------|--------|-------|--------|--------|
| I love co-activation graph | 0.025  | -0.15  | 0.05  | 0.125  | -0.15  |

Using this approach, the co-activation graph is created by constructing nodes representing each *hidden unit* and connecting them with edges signifying co-activation. For each sequence, we consider its average vector; when two *hidden units* consistently show similar activation patterns across sequences, they are connected in the graph. Thus, the co-activation graph summarizes dimension relationships within the Transformer's hidden layer. Calculating correlations between *hidden units* of encoder layers enables co-activation graph construction. Using the method in Section 2.3, correlation strengths across all encoders

form the graph's basis. Previously, a static threshold of 0.3 was used, considering only correlations above 0.3 significant. This empirical threshold is suboptimal since it impacts the graph's topology and may harm graph algorithms. Therefore, an optimal threshold strategy is needed for Transformers. The MASS meteric is useful here to set a more appropriate threshold Yan et al. (2018).

To determine the threshold with MASS, we leverage the spectral similarity between the original graph and a subgraph obtained by thresholding edge weights. This measure captures structural properties, like communities or core-periphery, remain stable under thresholding if weights and degrees correlate. MASS is computed using the largest eigenvalues of the Laplacian matrices of the original and difference graphs. We compute MASS for multiple thresholds from 0 to 1 to observe how the co-activation graph's structure evolves and identify the optimal threshold that balances retaining structure and simplifying complexity.

## 4. Experiments and Results

In order to assess whether the co-activation graph can help interpret how Transformer models function, we conducted two preliminary experiments: one focused on a Natural Language Processing (NLP) task, and the other on a Computer Vision task. This section introduces the datasets and outlines the experimental setup. Subsequently, we present two analyses.

In the first analysis, the co-activation graph is used to examine how relationships between nodes in the encoder layers and output classes can highlight dimensions that significantly influence predictions. This evaluates whether co-activation graphs reveal features contributing most to decision-making. In the second analysis, we apply a community detection algorithm to the co-activation graph to investigate whether the community structure reveals meaningful semantic groupings aligned with the model's learned representations. Together, these analyses evaluate the feasibility of using co-activation graphs to extract knowledge from trained Transformer models.

### 4.1. Datasets, Experiments, and Setup

For the NLP experiment, we used the 20-Newsgroups dataset, consisting of approximately 20,000 documents across 20 categories by Lang (1995). We employed two Transformer models: a custom-built Transformer (81.25% accuracy) and a fine-tuned *bert-base-uncased* model Devlin et al. (2019) (83.26% accuracy). For BERT, we extracted activations from the classification layer and the last two encoder layers.

For the vision experiment, we used the CIFAR-10 dataset by Krizhevsky and Hinton (2010), which includes 60,000 color images across 10 classes (50,000 training, 10,000 test). We fine-tuned the pre-trained *vit_base_patch16_224* Vision Transformer Wu et al. (2020), achieving 97.76% accuracy. For comparison, we used MobileNetV2 CNN Sandler et al. (2019) with reported accuracy of 94.43% Horta et al. (2021). Results are in Appendix B.

For both experiments, we constructed co-activation graphs by passing test samples through models, extracting activations, and computing Spearman correlations between hidden units. Using a threshold of 0 with the MASS metric, graphs were pruned. We applied Jaccard similarity and community detection to assess the structure and interpretability of the graphs.

## 4.2. Experiment with NLP task

### 4.2.1. Correlation between Classification Error and Jaccard Similarity

The primary objective of this subsection is to assess how effectively the co-activation graph captures class similarity derived from transformers. Our premise is that if a model frequently misclassifies a pair of classes, it suggests these classes are perceived as similar within its internal representation. To explore this, we investigate the correlation between classification error frequency and class similarity. We use the Jaccard Similarity coefficient, a metric for measuring similarity and diversity of sample sets. We hypothesize that more similar classes are likelier to be confused by the model, increasing classification errors.

The Jaccard coefficient in a graph is calculated as the size of the intersection of the sets of neighbors for two nodes divided by the size of the union of these sets; essentially, it measures the proportion of shared connections in the two nodes respective neighborhoods

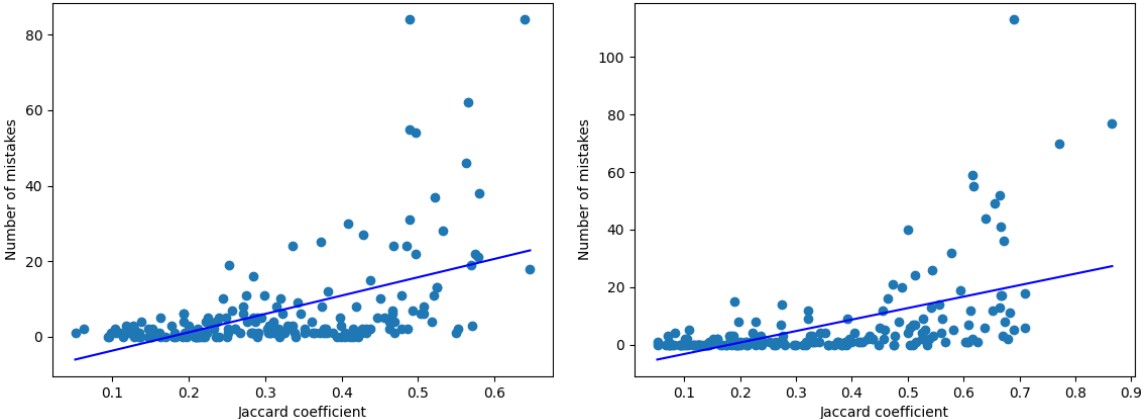

Figure 2: Correlation between jaccard coefficient and number of mistakes for the custom transformer (left) and the BERT model (right)

In our analysis, the custom model demonstrated a Spearman correlation coefficient of 0.526, indicating a moderate positive correlation between the Jaccard similarity and the count of classification errors. On the other hand, the BERT model displayed a higher correlation, with a Spearman coefficient of 0.634. This suggests a stronger relationship for the BERT model, implying that as the similarity between classes increases, there's a tendency for the model to make more classification errors. The difference in correlation values further emphasizes the distinct behavior of the two models.

While the traditional Jaccard coefficient doesn't consider the weights associated with edges in the network, our proposed modification treats membership from a probabilistic standpoint. This modification is highly fitting for our context, where edge weights signify the intensity of correlations between neurons.

In this probabilistically adjusted approach, we formulate the weighted Jaccard coefficient as follows:

$$J_w(a, b) = \frac{P(n(a) \cap n(b))}{P(n(a) \cup n(b))}$$

In this formula, $n(a)$ and $n(b)$ represent the sets of neighboring nodes for two given nodes $a$ and $b$, respectively. As an illustration, for a node $x \in n(a)$, instead of a binary membership as in the traditional Jaccard coefficient, we consider a probabilistic membership. This is determined by the edge weight, i.e., the correlation between nodes $a$ and $x$. Here is a more detailed formula for calculating the weighted Jaccard index:

$$J_w(a,b) = \frac{\sum_{x \in (n(a) \cap n(b))} w(x,a) \times w(x,b)}{\sum_{x \in n(a)} w(x,a) + \sum_{x \in n(b)} w(x,b) - \sum_{x \in (n(a) \cap n(b))} w(x,a) \times w(x,b)}$$

In this formula, $w(x,a)$ and $w(x,b)$ represent the weights of edges between the common neighbors $x$ and the nodes $a$ and $b$. For the denominator, we total the weights of edges for all neighbors of $a$ and $b$ individually, then subtract the sum of the weights of edges that connect common neighbors to both $a$ and $b$.

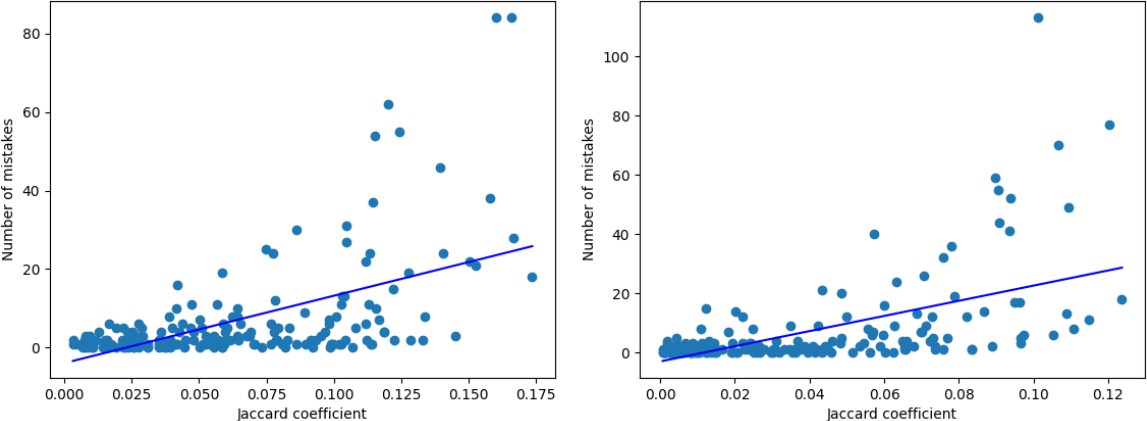

Figure 3: Correlation between weighted jaccard coefficient and number of mistakes with for the custom transformer (left) and the BERT model (right)

Upon employing our modified Jaccard coefficient, we obtained a Spearman correlations of 0.640 and 0.524 for the BERT and custom models, respectively. These coefficients bear notable similarity to those obtained previously using the traditional Jaccard formula.

However, both methodologies yield differing results in the Jaccard calculations. As per the first approach, the most overlapping nodes yield a Jaccard index of 0.65 for the custom model, while the BERT model presents a higher score of approximately 0.85. Conversely, in the weighted consideration method, the scenario is reversed. The Jaccard index for the most overlapping nodes is considerably lower at 0.15 for the BERT model and slightly higher at 0.175 for the custom model. This disparity underscores the influence of the weight consideration method in the determination of overlap.

In essence, our research demonstrated a significant positive correlation between class similarity and the number of mistakes involving them. This leads us to analyze communities and node similarities in the co-activation graph, we can glean insights into how Transformers perceive class similarities. Moreover, it suggests that nodes overlapping between two classes could be prime targets for tuning, enhancing the model's ability to distinguish between those classes. Nonetheless, while our emphasis is on assessing the fidelity of the graph

representation, a detailed evaluation of this particular aspect remains outside the purview of this study.

### 4.2.2. Community Analysis

The 20-newsgroup dataset by Lang (1995) comprises 20 categories that intuitively fall into broader groups such as Computer Technology, Recreational Activity, Science, Miscellaneous, Politics, and Religion. To show that co-activation graphs extract associated knowledge from Transformer models, two community detection algorithms were used. Results from the Louvain algorithm Blondel et al. (2008) appear in Tables 3 and 4. The Leiden algorithm Traag et al. (2019), with varying resolution parameter gamma, tracked community evolution; results are shown in Figure 4. These methods output a modularity coefficient to check how community structure differs from random graphs. Modularity ranges from [-1,1], with higher values indicating stronger connections within communities than between them. Notably, for

Table 3: Classes and their communities for the custom model. Modularity: 0.31

| Classes | Community |
|---------|-----------|
| C1 | alt.atheism, sci.crypt, soc.religion.christian, talk.politics.mideast, talk.politics.misc |
| C2 | comp.graphics, comp.os.ms-windows.misc, comp.sys.ibm.pc.hardware, comp.sys.mac.hardware, comp.windows.x, misc.forsale, sci.electronics |
| C3 | rec.autos, rec.motorcycles, rec.sport.baseball, rec.sport.hockey, sci.med, sci.space, talk.politics.guns |

the BERT model, community grouping was unstable due to the Louvain algorithm's non-deterministic nature. To improve reliability, we ran the algorithms 1000 times, calculating the frequency of each community grouping. One grouping was dominant with frequency 0.498, while others were below 0.225. Table 4 shows the result with the best frequency. The same approach was applied to the Leiden algorithm (see Figure 4), retaining only the most frequent groupings.

Table 4: Classes and their communities for the BERT model. Modularity: 0.33

| Classes | Community |
|---------|-----------|
| C1 | rec.sport.baseball, rec.sport.hockey |
| C2 | alt.atheism, rec.motorcycles, sci.crypt, sci.med, sci.space, soc.religion.christian, talk.politics.guns, talk.politics.mideast, talk.politics.misc, talk.religion.misc |
| C3 | comp.os.ms-windows.misc, comp.graphics, comp.sys.ibm.pc.hardware, comp.sys.mac.hardware, comp.windows.x, misc.forsale, sci.electronics |

Tables 3 and 4 show community detection for custom and BERT models, grouping 20 Newsgroups categories. In Table 3, the custom model clusters classes by content: C1 covers ideological discourse, C2 technologies, and C3 recreational and scientific topics. In Table 4, BERT groups by theme: C1 sports, C2 technologies, and C3 ideological discourse plus science.

Interestingly, *sci.crypt* is in Community C1 in the custom model. While one might expect it near C2 or C3 (science and technology), this suggests the model sees *sci.crypt* closer to ideological discourse. Possibly, cryptography articles in this dataset are framed in religious and political contexts.

Table 5: Groups of classes and their most important words

| Group of classes | 15 most important words based on tf-idf |
|---|---|
| alt.atheism, soc.religion.christian, talk.politics.mideast, talk.politics.misc, talk.religion.misc | writes, subject, article, organization, lines, would, one, people, god, university, dont, think, know, like, say |
| comp.os-ms-windows.misc, comp.sys.ibm.pc.hardware, comp.sys.mac.hardware, comp.windows.x, misc.forsale, sci.electronics | lines, subject, organization, university, nntp-postinghost, would, one, windows, know, thanks, writes, get, use, like, distribution |
| rec.autos, rec.motorcycles, rec.sport.baseball, rec.sport.hockey, sci.med, sci.space, talk.politics.guns | subject, organization, lines, writes, article, would, university, one, nntppostinghost, like, dont, get, know, think, im |
| sci.crypt | clipper, key, would, chip, writes, subject, lines, organization, encryption, article, one, government, nntppostinghost, use, people |

Table 5 shows the 15 most significant words per class group according to the TF-IDF score. *sci.crypt* shares many terms with Community C1, such as writing, subject, lines, organization, article, and people. It also contains politically related words such as government, which may influence clustering with C1. This suggests a dataset bias where cryptography articles often discuss political topics.

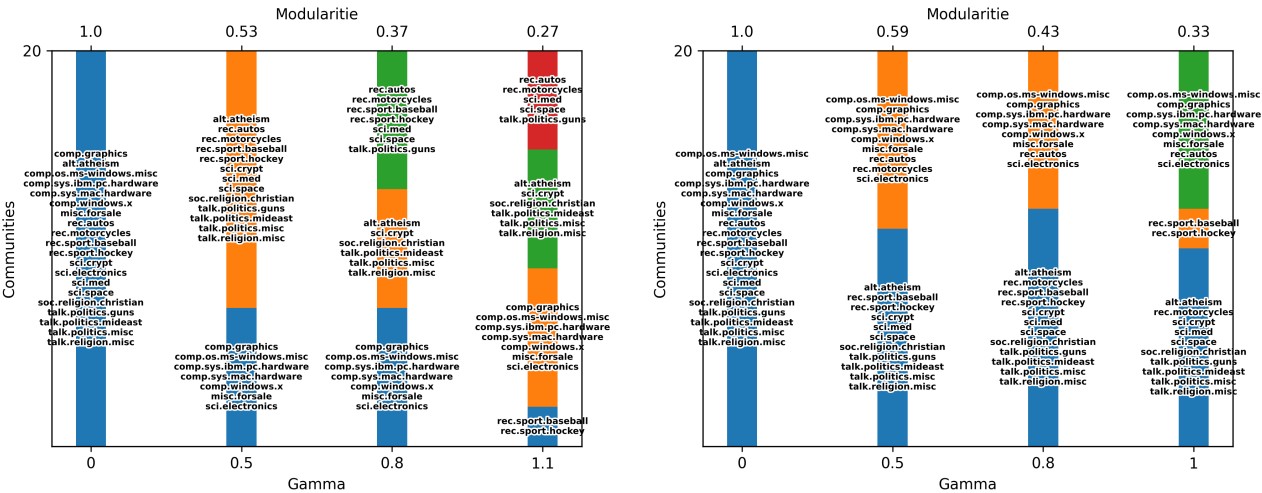

Figure 4: Leiden Communities with Gamma Variations for the custom transformer (left) and BERT model (right)

With the Leiden algorithm, increasing gamma sharpens community distinctions. As gamma increases, modularity decreases, reflecting weaker overall division due to the formation of smaller, more specific communities. This analysis reveals how models interpret and classify the 20 Newsgroups dataset at varying levels of granularity. Gamma affects BERT and the custom model differently. BERT's community structure remains stable across gamma values, while the custom model exhibits greater variation. This suggests that BERT's final encoder layers capture broader, more general knowledge, whereas the smaller custom model encodes a narrower but more discriminative representation. Our results show that community analysis via co-activation graphs is a powerful tool for extracting knowledge from Transformers. It highlights the effects of data bias and architectural differences. In the next phase, we will explore knowledge partitioning in Transformers by analyzing node centrality.

## 5. Limitations

While our approach provides a novel framework for interpreting Transformer representations using co-activation graphs, a few limitations remain. The use of mean pooling for dimensionality reduction may omit finer contextual signals. The evaluation is currently restricted to a limited set of models and datasets.. Additionally, MASS-based thresholding introduces computational cost and requires empirical calibration. Community detection outcomes are also sensitive to algorithmic choices such as resolution parameters. Ongoing work is addressing these aspects with new datasets, models, and improved experimental design, which will be reported in forthcoming publications.

## 6. Conclusion

In this paper, we presented a novel method for extracting knowledge from encoder-only Transformer models using co-activation graphs and graph analysis. We adapted the co-activation graph methodology, originally designed for CNNs, to Transformer architectures, and demonstrated its feasibility and effectiveness on both NLP and vision tasks. We conducted experiments involving a custom-built Transformer and a fine-tuned BERT model for NLP classification, with additional experiments on a pre-trained Vision Transformer (ViT) model exploring cross-domain applicability. We applied graph-based techniques such as community detection and weighted Jaccard similarity to analyze the co-activation graphs and reveal insights into the internal workings of Transformer models. Our results demonstrate a positive correlation between class similarity and classification errors, while community detection successfully identified semantically meaningful class clusters that align with intuitive domain groupings. These findings suggest that co-activation graphs can capture class similarities, semantic groupings, and structural patterns in a meaningful and interpretable way. While these are preliminary results with acknowledged limitations in experimental scope and computational scalability, they demonstrate the potential of using co-activation graphs to extract structured knowledge from Transformer models. Future work will investigate node centrality analysis to understand knowledge distribution across layers, validation on broader datasets including modern benchmarks, and integration with complementary explainability methods such as attention-based visualizations to produce more comprehensive explanations for Transformer behavior.

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

## Appendix A.  Flowchart of the co-activation graph for Transformers

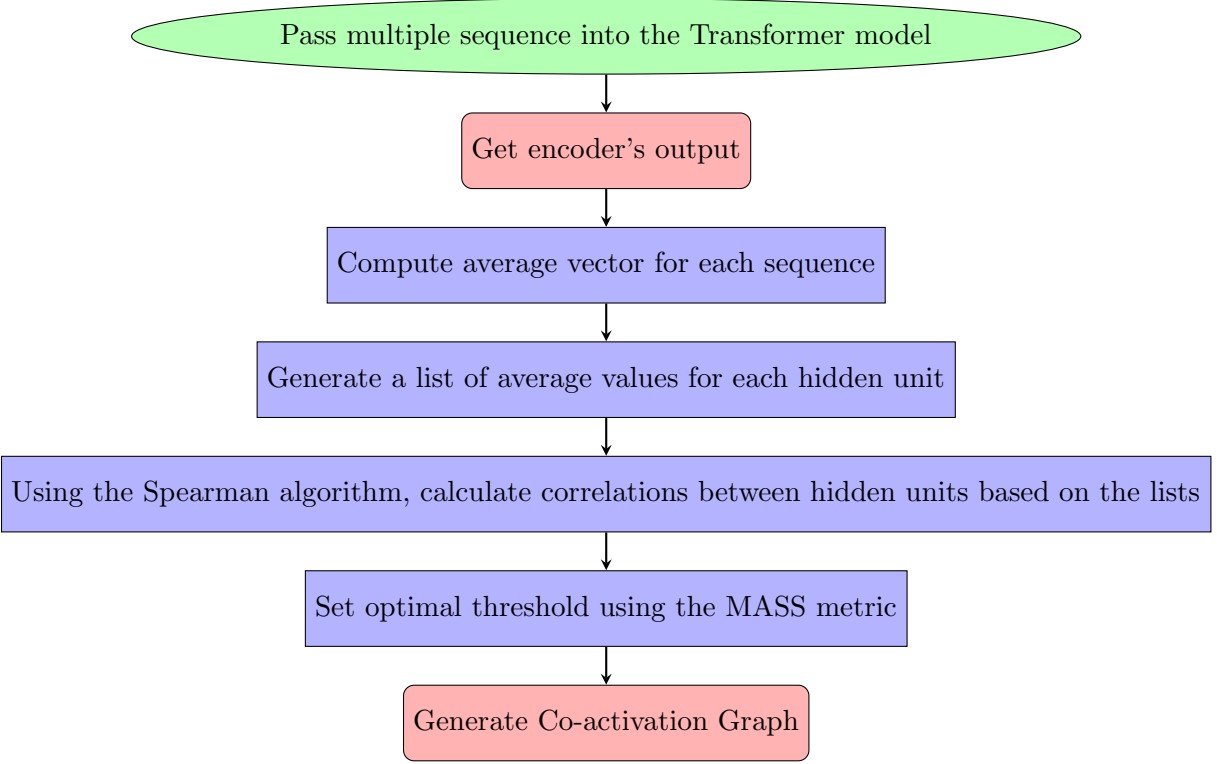

Figure 5: Flowchart of the co-activation graph generation methodology for Transformers with a data node before the encoder.

## Appendix B.  Experiments on Vision Tasks

As an additional experiment, we compare community analysis results from CNNs and a Transformer model designed for vision tasks—the Vision Transformer (ViT). This comparison aims to demonstrate that co-activation graphs can be effectively applied to ViT architectures as well.

### B.1.  Community Analysis

The CIFAR-10 dataset, which consists of ten distinct classes, can be intuitively divided into two main clusters: animals and transportation. We first present the groupings identified by the CNN model to determine whether the ViT model exhibits similar clustering behavior. As shown in Table 6, the CNN effectively separates these two groups.

| Community | Classes |
|-----------|---------|
| C1 | Deer, Dog, Horse |
| C2 | Frog, Bird, Cat |
| C3 | Airplane, Ship, Truck, Automobile |

Table 6: Community composition in MobileNetV2 model applied to the CIFAR-10 dataset

| Community | Classes |
|-----------|---------|
| C1 | Bird, Deer, Cat, Dog, Frog, Horse |
| C2 | Automobile, Airplane, Ship, Truck |

Table 7: Community composition in ViT model (vit_base_patch16_224) applied to the CIFAR-10 dataset

As shown in Table 7, the ViT model's community structure closely aligns with that of the CNN. Community C1 includes all animal-related classes, while C2 contains the transportation-related ones. This result supports the applicability of co-activation graph analysis to Transformer models in vision tasks and shows that similar class-level structures can be identified across different architectures.

