# OpenReview forum: "Adapting Graph-Based Analysis for Knowledge Extraction from Transformer Models"
_nesyconf.org/NeSy/2025/Conference_Phase_2 — NeSy 2025 - Phase 2 Poster_

### Official Review · Reviewer_keLN · 2025-07-03
**NeSy 2025 paper 89 review**

**Rating:** 8
**Confidence:** 5

**Review:**

The paper ‘Adapting Graph-Based Analysis for Knowledge Extraction from Transformer Models’ extends the method of extracting knowledge through graph-based representation from CNNs to Transformer architecture. The main contributions of this work are methods extensions and extensive experimental evaluations on NLP and computer vision tasks unveiling correlation between class similarity and classification errors and identification of semantically similar clusters. This is an interesting work contributing to explainability of transformers.

Major comments:

1. In Section 2.3 ‘Co-activation Graphs for CNNs’ one of the core concepts of the paper, a co-activation graph is mentioned. It would be good to provide a more formal definition of this notion.

2. In Section 3 ‘Generating co-activation graphs for transformers’ community analysis is mentioned. For better understanding it would be good to elaborate more on this notion: what is this, how it can be applied etc.

3. In Section 3 ‘Generating co-activation graphs for transformers’ we read ‘...the co-activation graph is created...’ – it would be good for easier understanding of what a co-activation graph is include an example how it is constructed (how weights for edges are calculated, how neurons become nodes etc.), maybe based on illustrations from Tables 1 and 2.

4. In Section 3 ‘Generating co-activation graphs for transformers’ the MASS metric is mentioned; it needs to be introduced explicitly here for easier understanding of the content. Also, it would be good to add an example illustrating how this metric is calculated and what do the outputs mean.

5. For the sake of reproducability, it would be good to add (e.g., to the Appendix) a list of best thresholds mentioned in Section 3 ‘Generating co-activation graphs for transformers’ for computing MASS metric.

6. In Section 4.1 ‘Datasets, Experiments, and Setup’ citations for 20-Newsgroups dataset:

Lang, Ken. "Newsweeder: Learning to filter netnews." Machine learning proceedings 1995. Morgan Kaufmann, 1995. 331-339.

And CIFAR-10 dataset:

https://www.cs.toronto.edu/~kriz/cifar.html

need to be added.

7. In Section 4.2.1 ‘Correlation between Classification Error and Jaccard Similarity’ there’s a mention of totaling ‘...the weights of edges for all neighbours of $a$ and $b$ individually...’, yet in the formula there’s a summation over elements of $n(a)$: $\sum_{x \in n(a)} x$; should there be $\sum_{x \in n(a)} w(x, a)$ instead? And the same for $n(b)$.

8. In Section 4.2.1 ‘Correlation between Classification Error and Jaccard Similarity’ it would be interesting to try to explain why correlation values for the custom transformer are moderate.

9. In Section 4.2.1 ‘Correlation between Classification Error and Jaccard Similarity’ a ‘significant positive correlation’ (p. 7) is mentioned; yet generally this indicated by values above 0.7; it might be better to write about ‘moderate positive correlation’ instead.

10. In Section 4.2.2 ‘Community Analysis’ it is not quite clear from the text how analyses reported were evaluated; was it done by modularity scores of two algorithms described in this section?

11. In Section 4.2.2 ‘Community Analysis’ citations are missed for Louvain algorithm:

Blondel, Vincent D., et al. "Fast unfolding of communities in large networks." Journal of statistical mechanics: theory and experiment 2008.10 (2008): P10008.

And for Leiden algorithm:

Traag, Vincent A., Ludo Waltman, and Nees Jan Van Eck. "From Louvain to Leiden: guaranteeing well-connected communities." Scientific reports 9.1 (2019): 1-12.

12. In Section 4.2.2 ‘Community Analysis’, in the Leiden algorithm, how the parameter gamma was chosen?

13. In Section 4.2.2 ‘Community Analysis’ two algorithms need to be introduced (at least) briefly for easier understanding.

14. Some words reported in Table 5 are not directly related by meaning to groups of classes they are associated with (e.g., ‘would’ or ‘one’ are more related to grammar, ‘nntppostinghost’ is probably more related to the information source). Have you tried to extract most important words by meaning? They could probably give more insights into the underlined groups of classes.

15. In Section 4.2.2 ‘Community Analysis’ there are some remarks regarding how the parameter gamma affects modularity in case of Leiden algorithm. It would be interesting to add some experimental results demonstrating this (maybe to the Appendix).

16. For the sake of reproducibility, it would be good to add code implementation and share the data.

Minor comments:

1. For easier readability it would be good to add brackets to cited works (e.g., (Vaswani et al. (2017)) on page 1 etc.).

2. In Section 3 ‘Generating co-activation graphs for transformers’ Table 2 needs to be mentioned as a reference, not Table 1.

3. For easier readability it is better to add separate captions to each subfigure in Figures 2 and 3 (e.g., for Figure 2, general caption: ‘Correlation between jaccard coefficient and number of mistakes’, left subfigure caption ‘Custom transformer’, right subfigure caption ‘BERT model’).

4. For easier understanding it would be better to place Figure 4 closer to paragraph 1 of Section 4.2.2 ‘Community Analysis’ since it is mentioned there.

**Anonymity:**

Remain anonymous

---

### Official Review · Reviewer_LaYK · 2025-07-04
**Promising Direction, Limited Analysis**

**Rating:** 4
**Confidence:** 4

**Review:**

This paper adapts a graph-based knowledge extraction technique, originally developed for CNNs, to Transformer architectures. By constructing co-activation graphs from encoder layers, where nodes represent hidden units and edges reflect activation correlations, the authors analyze the internal structure of Transformer classifiers. Experiments with a custom Transformer and BERT on NLP tasks reveal that semantically similar classes are more prone to mutual misclassification. The findings suggest that co-activation graphs offer a promising framework for uncovering interpretable and structured representations within Transformer models.

Pros:

* The paper explores a very interesting direction, representing Transformer internals using co-activation graphs.
* The idea of adapting graph-related algorithms (e.g. link prediction, community detection) in the context of model interpretability feels novel and compelling.

Cons:

* I found the paper hard to read at times (please refer to the following comments);
* The experimental findings, particularly the idea that similar classes (in terms of neuron activation correlations) are more likely to be confused, are somewhat expected and fail to deliver deeper insight from the graph-based representations; for instance, causal insights (e.g., via ablations or perturbations) would strengthen claims about learned structure and neuron function.




Abstract:

* I found this sentence a bit ambiguous: "Building upon the previous work on extracting knowledge from trained Transformer models
through a graph-based representation for Convolutional Neural Networks (CNN), we adapt the methodology to Transformer architectures". Please consider rephrasing it.

Introduction:

* Rephrase "This work builds upon prior CNN-based co-activation graphs and extends the approach to .." to something like "This work builds upon prior approaches based on CNN-based co-activation graphs..";
* "(ii) Evidence of a positive correlation between class similarity and classification errors. (iii) Graph-based identification of
semantically similar class clusters and their impact on misclassification.":  what is class similarity in this context? What classification are the authors referring? Are the authors testing on a classification task? At this point of reading, it was unclear.

Related Work:

* "They found that interpretable, specialized heads are pruned last...": I felt like this sentence required some additional details. What is pruning in this context? Why are specialized heads pruned?
* Related work seems a bit outdated. I suggest incorporating recent literature related to transformer intepretability, and importance attribution, in particular recent papers related to mechanistic approaches and circuit discovery. For instance, a work by Marks et al. (ICLR 2025) [1] and Garcia-Carrasco et al. (AAAI 2025) [2] seem to be very relevant.
* I found Table 1 a bit unnecessary.

Section 3:

* "The encoder’s output matrix corresponds directly to the input sequence, so each
element has a context tied to its input token...": While I understand the point, I think this entire paragraph has some awkward phrasing and it is difficult to comprehend.
* "An effective technique to calculate correlation between these vectors is reducing their dimensionality to a single value via an operation that retains significant information": this sentence is potentially ambiguous. Clarify whether the authors are averaging over tokens or dimensions. As written, it could be interpreted either way, although I understand the average is token-wise.
* "Previously, a static threshold of 0.3 was used, considering only correlations above 0.3 significant.". The term "previously" lacks a clear referent. Does this refer to previous literature, a baseline, or earlier experiments in this paper?
* "Table 1 illustrates how to compute the average vector for a sequence". Table 1 doesn't really show this, as it only reports example values for 5 dimensions. Maybe Table 1 and 2 could be merged into a single table for clarity.

Section 4:

* I would restructure the introductory part of the section to reflect the order of the following subsections. The first paragraph concludes with " Subsequently, we present two analyses.". However, the section goes on introducing the two analysis, while it moves to the datasets only later.
* Consider expanding this "For both experiments, we constructed co-activation graphs by passing test samples through models, extracting activations, and computing Spearman correlations between hidden units.." with explicit formulas.
* I found section 4.2 especially challenging to read because figures (e.g. Figure 2) present results before the metrics are defined. I think it would greatly benefit readability if the authors introduced metrics before section 4.2 (e.g. the weighted Jaccard coefficient in section 4.2).
* Consider making the class name in Table 3 and 4 explicit (e.g. "technologies" for C2).

Additional Comments and Typos:

* Missing spaces after periods are frequent;
* The paper uses the term "classes" ambiguously (at least before Section 4). It is unclear whether "class" refers to dataset labels, neuron clusters, or groups based on co-activation.

Overall Evaluation:

I think the paper starts from a very compelling idea, but ultimately the experimental contributions currently fall short of delivering substantial new insights. The core hypothesis, that frequent misclassification correlates with semantic similarity between classes, is intuitive and has limited novelty on its own. I think this betrays the premise of the study, as representing the inner neurons and interactions in a graph structure seems like a very interesting idea, more effort could be dedicated to actual analysis. I liked the community detection analysis but I think it could be very data-dependent, meaning that it is unclear whether co-activation graphs are uncovering patterns learned by the model, or simply surfacing statistical regularities in the input data. At present, I am leaning towards a rejection, though I acknowledge the potential for this line of work with further refinement.


[1] Marks, S., Rager, C., Michaud, E. J., Belinkov, Y., Bau, D., & Mueller, A. (2025). Sparse Feature Circuits: Discovering and Editing Interpretable Causal Graphs in Language Models. ICLR.

[2] Garcı́a-Carrasco, J., Maté, A., & Trujillo, J. (2025). Extracting Interpretable Task-Specific Circuits from Large Language Models for Faster Inference. AAAI, 16772–16780.

**Anonymity:**

Remain anonymous

---

### Official Review · Reviewer_meK1 · 2025-07-07
**Co-activation graphs, which were first used with CNNs, are further extended to Transformer models in this paper.   The authors construct graphs with nodes representing encoder-layer hidden elements and edges representing correlations between activation levels. The purpose of these co-activation graphs is to identify new groups within the inner representations of the model, examine semantic class similarity, and provide a justification for misclassifications. These graphs are in line with the observed patterns of conflict between semantically related classes, according to experiments conducted on a custom Transformer and BERT. Given the lack of a code repository in the manuscript, restricting the ability to go through documentation, results and reproductability along with the evaluation on a small number of datasets and similarity metrics, I grade the paper with a 6: Marginally above acceptance threshold.**

**Rating:** 6
**Confidence:** 4

**Review:**

### Quality:

* Although the method is conceptually interesting, the experimental scope appears to be limited, and the pipeline for graph building is only partially defined. The results are promising, however they are based on a single dataset for each experiment, which lowers the methodology's trustworthiness.

### Clarity:

* The paper is well-written and flows logically, making it easy to follow. The visual pipeline is very helpfull in understanding the Transformer architecture. However, more elaboration as to how this methodology advances from related effords would strenghten the overall showcase.

### Originality:

* In order to optimize graph formation, this research uses Spearman correlations and the MASS metric to adapt co-activation graphs from CNNs to Transformer encoders in an innovative manner.


### Significance:

* Though the article specifically notes plans to integrate their co-activation graph method with other explainability approaches, such as attention-based visualizations, indicating awareness of the need for complementary analysis, the method's practical utility remains uncertain without broader validation and clearer comparisons to existing tools. This could prove useful as a complementary tool for understanding model behavior, especially for classification tasks.

### Pros:

* By extending co-activation graphs, which were previously utilized for CNNs, to Transformers, the work fills an accessibility gap.
 In comparison to the tested Transformer architectures, how could this method work on larger or more diverse Transformer architectures?

 * Class similarity and classification error correlations that are positive imply that the graphs reflect relevant internal structure.

 * Visualizations are graph give a great insight along with the defined equations.

### Cons:

* Reproducibility and external validation are limited by the lack of publicly available code or full implementation instructions.

* Experiments cover only a few datasets/models, so generalization to other architectures or domains is uncertain. It would be usefull to see a comparison of datasets for the same parameters that would further enhance the overall usability.

* With no ablation or comparison to alternatives, the MASS metric's application for thresholding co-activation graphs is only briefly justified, leaving its impact on graph structure and interpretability unclear. A further explanation about result sensitivity could be beneficial.

* The metrics used, have limited scope and may not fully capture complex relationships in the data; future work should consider alternative similarity measures like cosine similarity, mutual information, or graph embedding-based metrics to provide a more comprehensive analysis.

**Anonymity:**

Remain anonymous

---

### Official Review · Reviewer_Pt2m · 2025-07-11
**Adapting Graph-Based Analysis for Knowledge Extraction from Transformer Models**

**Rating:** 7
**Confidence:** 5

**Review:**

The paper presents a novel adaptation of co-activation graph analysis from CNNs to Transformer models, offering a robust method to extract interpretable knowledge from complex architectures like BERT and ViT. Its key contributions include demonstrating a positive correlation between class similarity and classification errors, and using community detection to reveal semantic class clusters, enhancing understanding of Transformer behavior across NLP and vision tasks. This approach significantly advances model interpretability by providing structured, global insights into internal representations.

Comments:
1. Add some more up-to-date related works on Transformer interpretability and graph-based analysis (e.g., attention visualization or LLM graph methods).
2. Include the co-activation graph flowchart (Appendix A, Page 13) in Section 3 (Pages 4-5) to clarify the methodology and remove it from the appendix to avoid redundancy.
3. The paper does not explicitly address limitations, such as the potential loss of information from reducing activation vectors to mean values (Section 3, Page 4) or the computational cost of running community detection algorithms 1000 times to stabilize results (Section 4.2.2, Page 8). A dedicated limitations section would strengthen the paper by acknowledging these challenges upfront.
4. Move ViT results from the appendix to the main text to emphasize cross-domain applicability.

**Anonymity:**

Remain anonymous

---

### Official Review · Reviewer_umvQ · 2025-07-11
**Promising direction, but unclear methodology and weak validation**

**Rating:** 5
**Confidence:** 3

**Review:**

The paper proposes a graph-based co-activation method applied to the final layer outputs of BERT, treating each hidden unit as a graph node. Edges are constructed based on co-activation, followed by community analysis to cluster the nodes, resulting in a coarse-grained, posterior semantic analysis.

Strong Points:
	1. The paper presents a novel perspective. The proposed method demonstrates cross-domain transferability (NLP and CV), offering a partial explanation of the model’s behavior in the representation space.
	2. The paper introduces a weighted Jaccard index to validate the semantic plausibility of the edge construction strategy.

Weaknesses:
	1. Ambiguous architectural scope:
	- The paper uses both a "custom-built Transformer" and a "pre-trained BERT" for NLP tasks, and no clue for CV in abstract, but does not clarify in the main text that the paper focuses exclusively on encoder-only architectures. Please explicitly state in the Introduction that the proposed method is intended for encoder-only models such as BERT and ViT, to clarify the scope of the study.
	2. Lack of contrastive analysis:
	- The paper discusses encoder-only models in both NLP (BERT-like) and CV (ViT) domains, but does not discuss the potential effects of encoding differences between CV and NLP domains. Appendix B merely presents the results of applying the proposed framework to a CV task, without analysis.
	- Although both Louvain and Leiden algorithms are mentioned for community detection, the paper lacks any discussion or comparison of their performance across different models or tasks.
	- The paper lacks a discussion of the limitations and implementation details of the proposed pipeline.
	3. Limited validation in the NLP domain:
	- While emphasizing cross-domain transferability, the paper does not sufficiently validate its method within a single domain. In the NLP case, only the 20-Newsgroups dataset is used—a dated dataset from 1995 with just 20 categories—raising concerns about the method’s generalizability to modern semantic tasks.
	- The paper fails to properly cite the 20-Newsgroups dataset (e.g., see [1]), and does not specify which version of the dataset is used. Furthermore, it incorrectly categorizes the work of Xu et al. in Section 2 as Transformer-based, which it is not.
	4. Low readability and structural inconsistencies:
	- Section 2 includes a lengthy explanation of Transformer fundamentals that are only loosely related to the paper’s actual method.
	- The methodology is not presented as a standalone section. Instead, the description of the model is tightly coupled with the experimental section, making the paper read more like an experimental report than a theory-driven research article.
	- There are inconsistencies in table references (e.g., Table 2 is incorrectly cited as Table 1 in the main text).
	- Figures 2 and 3 lack clear explanations of the visual elements (e.g., slashes and dots), and are not properly referenced in the text.

References:
[1] Lang, Ken. Newsweeder: Learning to filter netnews. In Machine Learning Proceedings 1995. Morgan Kaufmann, 1995. pp. 331–339.
[2] Kelvin Xu, Jimmy Ba, Ryan Kiros, Kyunghyun Cho, Aaron Courville, Ruslan Salakhutdinov, Richard Zemel, and Yoshua Bengio. Show, attend and tell: Neural image caption generation with visual attention, 2016.

**Anonymity:**

Remain anonymous